# Quantifying the drivers of surface ozone anomalies in the urban areas over the Qinghai-Tibet Plateau

Hao Yin [1, 2,★], Youwen Sun [1, 2,†,★], Justus Notholt [3], Mathias Palm [3], and Cheng Liu [2,4,5,6†]

[1] *Key Laboratory of Environmental Optics and Technology, Anhui Institute of Optics and Fine Mechanics, HFIPS, Chinese Academy of Sciences, Hefei 230031, China*
[2] *Department of Precision Machinery and Precision Instrumentation, University of Science and Technology of China, Hefei 230026, China*
[3] *University of Bremen, Institute of Environmental Physics, P. O. Box 330440, 28334 Bremen, Germany*
[4] *Anhui Province Key Laboratory of Polar Environment and Global Change, University of Science and Technology of China, Hefei 230026, China*
[5] *Center for Excellence in Regional Atmospheric Environment, Institute of Urban Environment, Chinese Academy of Sciences, Xiamen 361021, China*
[6] *Key Laboratory of Precision Scientific Instrumentation of Anhui Higher Education Institutes, University of Science and Technology of China, Hefei 230026, China*
[†]Corresponding authors.
[★]These authors contributed equally.
E-mail addresses: Youwen Sun (ywsun@aiofm.ac.cn) and Cheng Liu (chliu81@ustc.edu.cn)

**Abstract**

Improved knowledge of the chemistry and drivers of surface ozone over the Qinghai-Tibet Plateau (QTP) is significant for regulatory and control purposes in this high-altitude region in the Himalaya. In this study, we investigate the processes and drivers of surface ozone anomalies (defined as deviations of ozone levels relative to their seasonal means) between 2015 and 2020 in urban areas over the QTP. We separate quantitatively the contributions of anthropogenic emissions and meteorology to surface ozone anomalies by using the random forest (RF) machine learning model based meteorological normalization method. Diurnal and seasonal surface ozone anomalies over the QTP were mainly driven by meteorological conditions, such as temperature, planetary boundary layer height, surface incoming shortwave flux, downward transport velocity, and inter-annual anomalies were mainly driven by anthropogenic emission. Depending on region and measurement hour, diurnal surface ozone anomalies varied over -27.82 μg/m$^3$ to 37.11 μg/m$^3$, where meteorological and anthropogenic contributions varied over -33.88 μg/m$^3$ to 35.86 μg/m$^3$ and -4.32 μg/m$^3$ to 4.05 μg/m$^3$, respectively. Exceptional meteorology driven 97% of surface ozone nonattainment events from 2015 to 2020 in the urban areas over the QTP. Monthly averaged surface ozone anomalies varied with much smaller amplitudes than their diurnal anomalies, where meteorological and anthropogenic contributions varied over 7.63 μg/m$^3$ to 55.61 μg/m$^3$ and 3.67 μg/m$^3$ to 35.28 μg/m$^3$ from 2015 to 2020, respectively. The inter-annual trends of surface ozone anomalies in Ngari, Lhasa, Naqu, Qamdo, Diqing, Haixi and Guoluo can be attributed to anthropogenic emissions by 95.77%, 96.30%, 97.83%, 82.30%, 99.26%, and 87.85%, and meteorology by 4.23%, 3.70%, 2.17%, 3.19%, 0.74%, and 12.15%, respectively. The inter-annual trends of surface ozone in other cities were fully driven by anthropogenic emission, where the increasing inter-annual trends would have larger values if not for the favorable meteorological conditions. This study can not only improve our knowledge with respect to spatiotemporal

variability of surface ozone but also provides valuable implication for ozone mitigation over the QTP.

## 1. Introduction

The Qinghai-Tibet Plateau (QTP) (27-45° N, 70-105° E), with an average altitude of 4000m above sea level (a.s.l), is the highest plateau in the world. It is known as the "Roof of the World" and the "Third Pole" (Qiu, 2008;Yang et al., 2013;Yin et al., 2017). The QTP has an area of approximately $2.5\times10^6$ km$^2$ and accounts for about one quarter of China's territory (Duo et al., 2018). The QTP is the source region of five major rivers in Asia, i.e., the Indus, Ganges, Brahmaputra, Yangtze, and Yellow rivers, which provide water resource to more than 1.4 billion people (Immerzeel et al., 2010). The QTP has been verified to be a critical region for regulating Asian monsoon climate and hydrological cycle, and it is thus an important ecological barrier of the whole Asia (Loewen et al., 2007;Yanai et al., 1992). The QTP has long been regarded as a pristine region due to its low population and industrial levels (Zhu et al., 2013). Due to its unique features of landform, ecosystem and monsoon circulation pattern, the QTP has been regarded as a sensitive region to anthropogenic impact, and is referred to as an important indicator of regional and global climate change (Qiu, 2008). The exogenous and local atmospheric pollutants are potential to accelerate the melting of glaciers, damage air quality, water sources, and grasslands, and threaten climate on regional and global scales (Yin et al., 2017;Yin et al., 2019c;Sun et al., 2021d;Pu et al., 2007;Kang et al., 2016). Therefore, improved knowledge of the evolutions and drivers of atmospheric pollutants in the QTP is of great importance for understanding local ecological situation and formulating regulatory policies.

Surface ozone ($O_3$) is a major air pollutant that threatens human health and vegetation growth (Jerrett et al., 2009;Yin et al., 2021b). Surface ozone over the QTP is generated either from its local anthropogenic and natural precursors such as nitrogen oxides ($NO_x$), volatile organic compounds (VOCs), and carbon monoxide (CO) via a chain of photochemical reactions or transported from long-distance regions by downwelling from the stratosphere. Surface ozone level is sensitive to local emissions, and meteorological conditions and transport. Meteorological conditions affect surface ozone level indirectly through changes in natural emissions of its precursors or directly via changes in wet and dry removal, dilution, chemical reaction rates, and transport flux. Emissions of air pollutants affect surface ozone level by perturbing the abundances of hydroperoxyl ($HO_2$) and alkylperoxyl ($RO_2$) radicals which are the key atmospheric constituents in formation of ozone. Some previous studies have presented the variability and analyzed qualitatively the drivers of surface ozone over the QTP at a specific site or region (Xu et al., 2016;Yin et al., 2019b;Yin et al., 2017;Zhu et al., 2004). However, none of these studies have quantitatively separated the contributions of anthropogenic emission and meteorology. Separation of anthropogenic and meteorological drivers is very important since it conveys us exactly which processes drive the observed ozone anomaly and therefore right conclusions can be made on whether an emission mitigation policy is effective.

Chemical transport models (CTMs) are widely used to evaluate the influences of meteorology and anthropogenic emission on atmospheric pollution levels (Hou et al., 2022;Sun et al., 2021a;Yin et al., 2020;Yin et al., 2019a). However, there are significant uncertainties in the emission inventories and in the models themselves, and shutting down an emission inventory in CTMs may cause large nonlinear effect, which inevitably influences the accuracy, performance and efficient of CTMs (Vu et al., 2019;Zhang et al., 2020). Mathematical and statistical models such as the multiple

linear regression (MLR) model and general additive models (GAMs) have also been used in many studies to quantify the influence of meteorological factors (Li et al., 2019;Li et al., 2020;Yin et al., 2021a;Yin et al., 2022;Zhai et al., 2019).

Machine learning (ML) is a well-known field that has been developing rapidly in recent years. Machine learning is a fusion of statistics, data science, and computing which experiences use across a very wide range of applications (Grange et al., 2018). Unlike most ML models such as artificial neural networks which are hard to understand the working mechanisms, the random forest (RF) model is not a "black-box" method, its prediction process can be explained, investigated, and understood (Gardner and Dorling, 2001;Grange et al., 2018;Grange and Carslaw, 2019;Shi et al., 2021). Recently, RF model based meteorological normalization technique has been proposed and used to decouple the meteorological influence on atmospheric pollution. For example, Vu et al. (2019) have used this technique to demonstrate that the clean air action plan implemented in 2013 was highly effective in reducing the anthropogenic emissions and improving air quality in Beijing. Shi et al. (2021) have used this technique to quantitatively evaluate changes in ambient $NO_2$, ozone, and $PM_{2.5}$ concentrations arising from these emission changes in 11 cities globally during the COVID-19 lockdowns.

In this study, we investigate the evolutions, implications, and drivers of surface ozone anomalies (defined as deviations of ozone levels relative to their seasonal means) from 2015 to 2020 in the urban areas over the QTP. Compared with previous studies that focus on surface ozone over the QTP, this study involves in larger area and longer time span. Most importantly, this study separates quantitatively the contributions of anthropogenic emission and meteorology to surface ozone anomalies by using the RF model based meteorological normalization method. This study can not only improve our knowledge with respect to spatiotemporal variability of surface ozone but also provides valuable implication for ozone mitigation over the QTP. We introduce detailed descriptions of surface ozone and meteorological field dataset in section 2. The method for separating contributions of meteorology and anthropogenic emission is presented in section 3. Section 4 analyzes spatiotemporal variabilities of surface ozone from 2015 to 2020 in each city over the QTP. The performance of the RF model used for surface zone prediction over the QTP is evaluated in Section 5. We discuss the implications and the drivers of surface zone anomalies from 2015 to 2020 in each city over the QTP in section 6. We conclude this study in section 7.

## 2. Data sources

### 2.1 Surface ozone data

The QTP covers an area of 2.5 million square meter and has a population of around 3 million, with most of them living in several cities. During the in depth study of the atmospheric chemistry over the Tibetan Plateau, @Tibet field campaign, ozone photochemistry and its roles in ozone budget are of great interests in both background atmosphere and in QTP urban areas. The former represents the influence of anthropogenic emission and cross boundary transport on the nature cycle of ozone in pristine atmosphere. The latter represents not only the upper limit of ozone photochemistry contribution to its budget, also a demanding knowledge for the sake of ozone pollution management. Hourly surface ozone data in the urban areas over the QTP are available from the China National Environmental Monitoring Center (CNMEC) network (http://www.cnemc.cn/en/, last access: November 26, 2021). The CNMEC network based ozone measurements have been widely used in many studies for evaluation of regional atmospheric

pollution and transport over China (Lu et al., 2021;Lu et al., 2019a;Lu et al., 2020;Sun et al., 2021c;Sun et al., 2021d;Yin et al., 2021a;Yin et al., 2021b;Yin et al., 2022).The CNEMC network has deployed 33 measurement sites in 12 cities over the QTP (Table 1). The number of measurement sites in each city varies from 1 to 6. All surface ozone time series at each measurement site are provided by active differential absorption ultraviolet (UV) analyzers. For all the 33 measurement sites, hourly surface ozone data are available since 2015. We first removed unreliable measurements at all measurement sites in each city by using the filter criteria following our previous studies (Lu et al., 2018;Lu et al., 2020;Sun et al., 2021b;Sun et al., 2021d;Yin et al., 2021a;Yin et al., 2021b), then averaged all measurements in each city to generate a city representative dataset. All investigations in this study are performed on such city representative basis.

As illustrated in Figure 1, the QTP (Latitude range: 26°00' ~ 39°47', Longitude range: 73°19' ~ 104°47') covers the Kunlun Mountain, the A-erh-chin Mountain and the Qilian Mountain in the north, the Pamir Plateau and the Karakorum Mountains in the west, the Himalayas in the south, and the Qinling Mountains and the Loess Plateau in the east. These 12 cities are the most populated areas over the QTP. All these cities except Aba and Diqing are located in Tibet or Qinghai provinces. Aba and Diqing are in Sichuan and Yunnan provinces, respectively. The area of these cities ranges from 7.7 to 430 thousand $km^2$, the altitude ranges from 2.3 to 4.8 km a.s.l., and the population ranges from 0.12 to 2.47 million. The residents within the 12 cities are about 3.85 million account for about 51% of population over the QTP.

### 2.2 Meteorological data

Meteorological fields used in this study are from the Modern-Era Retrospective analysis for Research and Applications Version 2 (MERRA-2) dataset (Gelaro et al., 2017). The MERRA-2 dataset is produced by the NASA Global Modeling and Assimilation Office and it can provide time series of many meteorological variables with a spatial resolution of $0.5° \times 0.625°$ (The NASA Global Modeling and Assimilation Office (GMAO), 2022). The boundary layer height and surface meteorological variables are available per hour and other meteorological variables are available every 3 hours. It has been verified that the MERRA-2 meteorological fields over Chinese weather stations are in good agreement with the observations (Carvalho, 2019;Kishore Kumar et al., 2015;Song et al., 2018;Zhou et al., 2017). This MERRA-2 dataset has been extensively used in evaluations of regional atmospheric pollution formation and transport over China (Li et al., 2019;Li et al., 2020;Yin et al., 2022;Zhai et al., 2019).

## 3. Methodology

### 3.1 Quantifying seasonality and inter-annual variability

We quantify the seasonality and inter-annual variability of surface ozone from 2015 to 2020 in each city over the QTP by using a bootstrap resampling method. The principle of such bootstrap resampling method was described in detail in Gardiner et al. (2008). Many studies have verified the robustness of Gardiner's methodology in modeling the seasonality and inter-annual variabilities of a suite of atmospheric species (Sun et al., 2020;Sun et al., 2021a;Sun et al., 2021b;Sun et al., 2021d;Sun et al., 2018). In this study, we used a second Fourier series plus a linear function to fit surface ozone monthly mean time series from 2015 to 2020 over the QTP. The usage of measurements on monthly basis can improve the fitting correlation and lower the regression residual. As a result, the relationship between the measured and bootstrap resampled surface ozone monthly

mean time series can be expressed as,

$$V(t,\mathbf{b}) = b_0 + b_1 t + b_2 \cos\left(\frac{2\pi t}{12}\right) + b_3 \sin\left(\frac{2\pi t}{12}\right) + b_4 \cos\left(\frac{4\pi t}{12}\right) + b_5 \sin\left(\frac{4\pi t}{12}\right) \quad (1)$$

$$F(t,a,\mathbf{b}) = V(t,\mathbf{b}) + \varepsilon(t) \quad (2)$$

where $F(t,a,\mathbf{b})$ and $V(t,\mathbf{b})$ represent the measured and fitted surface ozone time series, respectively. The parameters $b_0 - b_5$ contained in the vector **b** are coefficients obtained from the bootstrap resampling regression with $V(t,\mathbf{b})$. The $b_0$ is the intercept, and the $b_1$ is the annual growth rate, and $b_1/b_0$ is the inter-annual trend discussed below. The parameters $b_2$–$b_5$ describe the seasonality, *t* is the measurement time in month elapsed since January 2015, and $\varepsilon(t)$ represents the residual between the measurements and the fitting results. The autocorrelation in the residual can increase the uncertainty in calculation of inter-annual trend. In this study, we have followed the procedure of Santer et al. (2008) and included the uncertainty arising from the autocorrelation in the residual.

### 3.2 Random Forest (RF) model

We have established a decision tree based random forest (RF) machine learning model to describe the relationships between hourly surface ozone concentrations (response variables) and their potential driving factors (predictive variables) in the urban areas over the QTP. As summarized in Table 2, predictive variables used in this study include time variables such as year 2015 to 2020, month 1 to 12, day of the year from 1 to 365, hour of the day from 0 to 23, and meteorological parameters such as wind, temperature, pressure, cloud fraction, rainfall, vertical transport, radiation and relative humidity. These time variables were selected as proxies for emissions since pollutant emissions vary by the time of day, day of the week, and season (Grange et al., 2018).

The detailed descriptions of RF machine learning model can be found in Breiman (2001). Briefly, the RF model is an ensemble model consisting of hundreds of individual decision tree models. Each individual decision tree model uses a bootstrap aggregating algorithm to randomly sample response variables and their predictive variables with a replacement from a training dataset. In this study, a single regression decision tree is grown in different decision rules based on the best fitting between surface ozone measurements and their predictive variables. The predictive variables are selected randomly to give the best split for each tree node. The predicted surface ozone concentrations are given by the final decision as the outcome of the weighted average of all individual decision trees. By averaging all predictions from bootstrap samples, the bagging process decreases variance and thus helps the model to minimize overfitting.

As shown in Figure 2, the whole dataset was randomly divided into (1) a training dataset to establish the random forest model and (2) a testing dataset (not included in model training) to evaluate the model performance. The training dataset was randomly selected from 70 % of the whole data and the remaining 30% was taken as the testing dataset. The hyperparameters for the RF model in this study were configured following those in Vu et al. (2019) and Shi et al. (2021) and are summarized as follows: the maximum tree of a forest is 300 (n_tree=300), the number of variables for splitting the decision tree is 4 (mtry=4), and the minimum size of terminal nodes is 3 (min_node_size=3). Since the meteorological variables differ in units and magnitudes, which could lead to unstable performance of the model. Therefore, we uniformized all meteorological variables via equation (3) before using them in the RF model. This pre-processing procedure can also speed up the establishment of the RF model.

$$\boldsymbol{z}_k = \frac{\boldsymbol{x}_k - \boldsymbol{u}_k}{\boldsymbol{\sigma}_k} \tag{3}$$

where $\boldsymbol{u}_k$ and $\boldsymbol{\sigma}_k$ are the average and 1σ standard deviation (STD) of $\boldsymbol{x}_k$, and $\boldsymbol{z}_k$ is the pre-processed value for parameter $\boldsymbol{x}_k$.

### 3.3 Separation of meteorological and anthropologic contributions

In order to separate the contributions of meteorology and anthropologic emission to surface ozone anomalies in each city over the QTP, we have decoupled meteorology driven anomalies by using the RF model based meteorological normalization method. The meteorological normalization method was first introduced by Grange et al. (2018) and improved by Vu et al. (2019) and Shi et al (2021). To decouple the meteorological influence, we first generated a new input dataset of predictive variables, which includes original time variables and resampled meteorological variables ($T_{surface}$, $U_{10}$, $V_{10}$, PBLH, CLDT, PRECTOT, OMEGA, SWGDN, QV, TROPH). Specifically, meteorological variables at a specific selected hour of a particular day in the input dataset were generated by randomly selecting from the meteorological data during 1980 to 2020 at that particular hour of different dates within a four-week period (i.e., 2 weeks before and 2 weeks after that selected date). For example, the new input meteorological data at 18:00, 15 February 2018, are randomly selected from the meteorological data at 18:00 on any date from 1 to 29 February of any year during 1980 to 2020. This selection process was repeated 1000 times to generate a final input dataset. The 1000 meteorological data were then fed to the RF model to predict surface zone concentration. The 1000 predicted zone concentrations were then averaged as equation (4) to calculate the final meteorological normalized concentration ($O_{3,\,dew}$) for that particular hour, day, and year.

$$O_{3,dew} = \frac{1}{1000}\sum_{i=1}^{1000} O_{3,i,pred} \tag{4}$$

where $O_{3,i,pred}$ is the surface ozone concentration predicted by using the $i^{th}$ meteorological data randomly selected from the meteorological data at the specific selected hour on any date from 1 to 29 February of any year in 1980 to 2020. $O_{3,\,dew}$ represents surface ozone concentration under the mean meteorological conditions at the specific selected hour between 1980 to 2020.

If the seasonal variabilities of anthropogenic emission and meteorology are constant over year, the variability of surface zone can be exactly reproduced by the seasonality plus the intercept in equation (1), i.e., the annual growth rate of surface ozone and the fitting residual should be close to zero. But this is not realistic in real world. Any year-to-year difference in either anthropogenic emission or meteorology could result in anomalies. We calculate surface ozone anomalies ($O_{3,anomalies}$) in each city over the QTP by subtracting their seasonal mean values ($O_{3,mean}$) from all hourly surface ozone measurements ($O_{3,individual}$) through equation (5) (Hakkarainen et al., 2019;Hakkarainen et al., 2016;Mustafa et al., 2021).

$$O_{3,anomalies} = O_{3,individual} - O_{3,mean} \tag{5}$$

where $O_{3,mean}$ in each city are approximated by the seasonality plus the intercept described in equation (1). As a result, the difference $O_{3,meteo}$ between $O_{3,individual}$ and $O_{3,dew}$ calculated as equation (6) is the portion of anomalies induced by changes in meteorology. The difference $O_{3,emis}$ between $O_{3,anomalies}$ and $O_{3,meteo}$ calculated as equation (7) represents the portion of anomalies induced by changes in anthropogenic emission.

$$O_{3,meteo} = O_{3,individual} - O_{3,dew} \tag{6}$$

$$O_{3,emis} = O_{3,anomalies} - O_{3,meteo} \tag{7}$$

Open Access EGU

By applying the meteorological normalization method, we finally separate the contributions of meteorology and anthropogenic emissions to the surface ozone anomalies in each city over the QTP. Positive $O_{3,meteo}$ and $O_{3,emis}$ indicate that changes in meteorology and anthropogenic emission cause surface ozone concentration above their seasonal mean values, respectively. Similarly, negative $O_{3,meteo}$ and $O_{3,emis}$ indicate that changes in meteorology and anthropogenic emission cause surface ozone concentration below their seasonal mean values, respectively.

## 4. Variabilities of surface ozone over the QTP

### 4.1 Overall ozone level

Statistical summary and box plot of surface ozone concentration (units: μg/m$^3$) in each city over the QTP from 2015 to 2020 are presented in Table S1 and Figure S1, respectively. The average of surface ozone between 2015 and 2020 in each city over the QTP varied over (50.67 ±29.57) μg/m$^3$ to (90.38 ± 28.83) μg/m$^3$, and the median value varied over 53.00 μg/m$^3$ to 90.00 μg/m$^3$. In comparison, the averages of surface ozone between 2015 and 2020 in the Beijing-Tianjin-Hebei (BTH), the Fenwei Plain (FWP), the Yangtze River Delta (YRD) and the Pearl River Delta (PRD) in densely populated and highly industrialized eastern China were 140.76 μg/m$^3$, 132.16 μg/m$^3$, 125.09 μg/m$^3$ and 119.82 μg/m$^3$, respectively. The average of surface ozone between 2011 and 2015 at the suburb Nam Co station in the southern-central of the QTP was (47.00 ± 12.43) μg/m$^3$ (Yin et al., 2019b). As a result, surface ozone levels in the urban areas over the QTP are much lower than those in urban areas in eastern China but higher than those in the suburb areas over the QTP. Among all cities over the QTP, the highest and lowest surface ozone concentration occurs in Haixi and Aba, with mean values of (90.38 ± 28.83) μg/m$^3$ and (50.67 ± 28.83) μg/m$^3$, respectively. Generally, surface ozone concentrations in Qinghai province are higher than those in Tibet province.

The ambient air quality standard issued by the Chinese government regularized that the critical value (Class 1 limit) for the maximum 8-hour average ozone level is 160 μg/m$^3$. With this rule, we summarize the number of nonattainment day per year in each city over the QTP in Table S1. The number of nonattainment day per city and per year over the QTP is only 2 between 2015 and 2020. Ozone nonattainment events over the QTP typically occur in spring or summer. In comparison, the number of nonattainment day per city and per year over the BTH, FWP, YRD and PRD are much larger, with values of 78, 36, 82 and 45 between 2015 and 2020, respectively, and all ozone nonattainment events over these regions occur in summer. The number of nonattainment day in Ngari in 2020, Lhasa in 2016 and 2017, Shannan in 2017 and 2018, Haixi in 2015 and 2019, and Xining in 2017 are 13, 10, 20, 12, 10, 14, 16, and 17 days, respectively. The number of nonattainment day in all other cities over the QTP are less than 10 days. Especially, surface ozone concentrations in Aba, Naqu, and Diqing in all years between 2015 and 2020 are less than the Class 1 limit of 160 μg/m$^3$. There are only 1 and 2 nonattainment days in Nyingchi and Qamdo between 2015 and 2020, respectively.

### 4.2 Diurnal variability

Diurnal cycles of surface ozone in each season and each city over the QTP are presented in Figure 3. Overall, diurnal cycle of surface ozone in each city over the QTP presents a unimodal pattern in all seasons. For all cities in all seasons, high levels of surface ozone occur in the daytime (9:00 to 20:00 LT) and low levels of surface ozone occur in the nighttime (21:00 to 08:00 LT). As seen from Figure 3, surface ozone levels usually increase over time starting at 8:00 to 11:00 LT in

the morning, reach the maximum values at 15:00 to 18:00 LT in the afternoon, and then decreases over time till the minimum values at 8:00 or 9:00 LT the next day.

The timings of the diurnal cycles in all cities over the QTP were shifted by 1 to 2 hours later in winter than those in the rest of the year, most likely due to the later time of sunrise. (Yin et al., 2017) also observed such shift in diurnal cycle at the suburb Nam Co station. The diurnal cycles of surface ozone in the urban areas over the QTP spanned a large range of −43.73 % to 47.12 % depending on region, season, and measurement time. The minimum and maximum surface ozone levels in the urban areas over the QTP varied over (22.89 ± 15.55) μg/m$^3$ to (68.96 ± 18.27) μg/m$^3$ and (57.77 ± 21.56) μg/m$^3$ to (102.08 ± 15.14) μg/m$^3$, respectively. On average, surface ozone levels in the urban areas over the QTP have mean values of (72.41 ± 33.83) μg/m$^3$ during the daytime (08:00-19:00) and (60.89 ± 32.25) μg/m$^3$ during the evening (20:00-08:00). The diurnal cycles of surface ozone in all cities over the QTP are generally consistent with the results reported in eastern China and the suburb areas over the QTP (Yin et al., 2019b;Yin et al., 2017;Zhao et al., 2016;Shen et al., 2014).

### 4.3 Seasonal variability

Monthly averaged time series of surface ozone in each city over the QTP between 2015 and 2020 are shown in Figure 4. Surface ozone levels in all cities over the QTP showed pronounced seasonal features. Seasonal cycles of surface ozone in most cities present a unimodal pattern with a seasonal peak occurs around March-July and a seasonal trough occurs around October-December. Specifically, maximum surface ozone levels occur in spring over Diqing, Lhasa, Naqu, Nyingchi, Qamdo, Shannan, Shigatse, Aba, and occur in summer over Ngari, Xining, Guoluo, and Haixi; Minimum surface ozone levels in Nyingchi and Diqing occur in autumn, and in other cities occur in winter. The minimum and maximum surface ozone levels between 2015 and 2020 over the QTP varied over (29.21 ± 19.03) μg/m$^3$ to (60.45 ± 31.35) μg/m$^3$ and (71.25 ± 26.53) μg/m$^3$ to (112.46 ± 28.92) μg/m$^3$, respectively (Table S2). The peak−to−trough contrast in Diqing, Naqu, Nyingchi, and Aba were smaller than those in other cities. Due to regional deference in meteorology and anthropogenic emission, seasonal cycle of surface ozone in the urban areas over the QTP is regional dependent.

### 4.4 Inter-annual variability

The inter-annual variability of surface ozone between 2015 and 2020 in each city over the QTP fitted by the bootstrap resampling method is presented in Figure 5 and summarized in Table S2. Generally, the measured and fitted surface ozone concentrations in each city over the QTP are in good agreement with a correlation coefficient (R) of 0.68–0.92 (Figure S2). The measured features in terms of seasonality and inter-annual variability can be reproduced by the bootstrap resampling model. However, due to the year-to-year deference in anthropogenic emission and meteorology, both inter-annual variability and fitting residual were not zero in all cities. The inter-annual trends in surface ozone level from 2015 to 2020 over the QTP spanned a large range of (−2.43 ± 0.56) μg/m$^3$·yr$^{-1}$ to (7.55 ± 1.61) μg/m$^3$·yr$^{-1}$, indicating a regional representation of each dataset. The inter-annual trends of surface ozone levels in most cities including Diqing, Naqu, Ngari, Nyingchi, Shannan, Shigatse, Xining, Abzhou and Haixi showed positive trends. The largest increasing trends were presented in Diqing and Nagri, with values of (5.31 ± 1.28) μg/m$^3$·yr$^{-1}$ and (7.55 ± 1.61) μg/m$^3$·yr$^{-1}$, respectively. In contrast, surface ozone levels in Lhasa, Qamdo and Guoluo presented negative trends, with values of (-1.62 ± 0.76) μg/m$^3$·yr$^{-1}$, (-2.43 ± 0.56) μg/m$^3$·yr$^{-1}$ and (-2.36 ± 0.81)

μg/m$^3$·yr$^{-1}$, respectively.

## 5. Performance evaluation

We evaluate the performance of the RF model in predicting hourly surface ozone level in each city over the QTP using the metrics of Pearson correlation coefficient (R), the root means square error (RMSE), and the mean absolute error (MAE). They are commonly used metrics for evaluation of machine learning model predictions, and are defined as equations (8), (9), and (10), respectively.

$$R = \frac{n\sum_{i=0}^{n} x_i y_i - \sum_{i=0}^{n} x_i \cdot \sum_{i=0}^{n} y_i}{\sqrt{n\sum_{i=0}^{n} x_i^2 - (\sum_{i=0}^{n} x_i)^2} \cdot \sqrt{n\sum_{i=0}^{n} y_i^2 - (\sum_{i=0}^{n} y_i)^2}} \tag{8}$$

$$RMSE = \sqrt{\frac{\sum_{i=1}^{n}(x_i - y_i)^2}{n}} \tag{9}$$

$$MAE = \frac{\sum_{i=1}^{n}|x_i - y_i|}{n} \tag{10}$$

where $x_i$ and $y_i$ are the $i^{th}$ concurrent measured and predicted data pairs, respectively. The *n* is the number of measurements. The *R* value represents the fitting correlation between the measurements and predictions. The RMSE value measures the relative average difference between the measurements and predictions. The MAE value measures the absolute average difference between the measurements and predictions. The units of RMSE and MAE are same as the measured data, namely μg/m$^3$.

Comparisons between the model predictions and measurements for the testing data (not included in model training) in each city over the QTP are shown in Figure S3. Overall, the RF model predictions and surface ozone measurements are in good agreements, showing high *R* and low RMSE and MAE for testing dataset in each city over the QTP (Figure S3). Depending on cities, the *R* values varied over 0.85 to 0.94, the RMSE over 10.24 to 17.55 μg/m$^3$, and MAE over 7.32 to 12.76 μg/m$^3$. The R, RMSE, and MAE are independent of city and surface ozone level. The results affirm that our model performs very well in predicting surface ozone levels and variabilities in each city over the QTP.

We also investigate the importance of each input variable in the RF model for predicting surface ozone level in each city over the QTP. As shown in Figure S4, time information such as hour term (Hour), year term (Year) or seasonal term (Month) are the most important variables in the RF model predictions in all cities except Xining and Haixi where temperature term ($T_{2m}$) is the most important variable. For all cities, the aggregate importance of time information is larger than 50%. In all cities over the QTP, the meteorological variables such as temperature ($T_{2m}$), relatively humidity (QV), Vertical pressure velocity (OMEGA) and Planetary boundary layer height (PBLH) play significant roles when explaining surface ozone concentrations. For other variables, although they are not decisive variables in the RF model predictions, they are not negligible in predicting surface ozone in all cities over the QTP. Although time information are the most important variables in the RF model predictions, they can be used very precisely, and thus the RF model to measurement discrepancy in all cities could be from other predictive variables rather than time information.

## 6. Drivers of surface ozone anomalies

### 6.1 Diurnal scale

Figure 6 presents diurnal cycles of surface ozone anomalies between 2015 and 2020 along with the meteorology-driven and anthropogenic-driven portions in each city over the QTP. In all cities,

the anthropogenic contributions are almost constant but the meteorological contributions show large variations throughout the day. Depending on region and measurement hour, diurnal surface ozone anomalies on average varied over -27.82 μg/m$^3$ to 37.11 μg/m$^3$ between 2015 and 2020, where meteorological and anthropogenic contributions varied over -33.88 μg/m$^3$ to 35.86 μg/m$^3$ and -4.32 μg/m$^3$ to 4.05 μg/m$^3$, respectively. The least contrast between meteorological contribution and anthropogenic contribution occurs in Haixi. The diurnal cycles of meteorological contribution are consistent with those of surface ozone anomalies. High levels of meteorological contribution occur in the daytime (9:00 to 20:00 LT) and low levels of meteorological contributions occur in the nighttime. As a result, diurnal surface ozone anomalies in each city over the QTP were mainly driven by meteorology.

We further investigated the drivers of surface ozone nonattainment events from 2015 to 2020 in each city over the QTP. All ozone nonattainment events were classified as meteorology-dominated or anthropogenic-dominated events according to which one has a larger contribution to the observed surface ozone nonattainment events. The statistical results are listed in Table S3. Except one day in Ngari in 2018, one day in Shigatse in 2016, and one day in Haixi in 2019 which were dominated by anthropogenic emission, all other surface ozone nonattainment events from 2015 to 2020 over the QTP were dominated by meteorology. Exceptional meteorology driven 97% of surface ozone nonattainment events from 2015 to 2020 in the urban areas over the QTP. For the meteorology-dominated surface ozone nonattainment events, meteorological and anthropogenic contributions varied over 32.85 μg/m$^3$ to 55.61 μg/m$^3$ and 3.67 μg/m$^3$ to 7.23 μg/m$^3$, respectively. For the anthropogenic-dominated surface ozone nonattainment events, meteorological and anthropogenic contributions varied over 7.63 μg/m$^3$ to 10.53 μg/m$^3$ and 15.63 μg/m$^3$ to 35.28 μg/m$^3$, respectively.

### 6.2 Seasonal scale

Figure 7 presents seasonal cycles of surface ozone anomalies between 2015 and 2020 along with the meteorology-driven and anthropogenic-driven portions in each city over the QTP. In all cities, the monthly averaged surface ozone anomalies between 2015 and 2020 varied with much smaller amplitudes than their diurnal anomalies. Noticeable anomalies include pronounced positive anomalies in December in Nagri, in May in Lhasa, Shannan, and Qamdo, in July in Haixi, in June in Guoluo, and negative anomalies in July in Lhasa, Nyingchi, and Guoluo. Both meteorological and anthropogenic contributions are regional dependent and show large variations throughout the year. Depending on region and month, meteorological and anthropogenic contributions varied over -4.54 μg/m$^3$ to 3.31 μg/m$^3$ and -2.67 μg/m$^3$ to 3.35 μg/m$^3$ between 2015 and 2020, respectively.

Seasonal surface ozone anomalies between 2015 and 2020 in all cities over the QTP were mainly driven by meteorology. For example, meteorology caused decrements of 3.05 μg/m$^3$ in July and 4.27 μg/m$^3$ in September in Diqing, while anthropogenic emission caused increments of 0.64 μg/m$^3$ and 1.34 μg/m$^3$ in respective months. Aggregately, we observed -2.41 μg/m$^3$ and -2.89 μg/m$^3$ of seasonal surface ozone anomalies in July and September in Ngari, respectively. In all cities, seasonal cycles of meteorological contribution are more consistent with those of surface ozone anomalies over the QTP. In some cases, surface ozone anomalies would have larger values if not for the unfavorable meteorological conditions, e.g., surface ozone anomalies in June in Ngari, in December in Shannan, Guoluo and Aba.

### 6.3 Annual scale

Annual mean surface ozone anomalies between 2015 and 2020 along with meteorology-driven and anthropogenic-driven portions in each city over the QTP are presented in Figure 8. Surface ozone in Diqing, Naqu, Nagri, Haixi and Shannan show larger year to year variations than those in other cities. Annual mean surface ozone levels in Diqing, Naqu, Nagri and Haixi showed significant reductions of 2.10 μg/m$^3$, 10.32 μg/m$^3$, 6.87 μg/m$^3$, and 15.97 μg/m$^3$, respectively, Shannan showed an increment of 9.12 μg/m$^3$, and other cities showed comparable values in 2016 relative to 2015. The largest year to year difference occurred in Ngari during 2016 to 2017, which has an increment of 25.25 μg/m$^3$. The results show that anthropogenic contributions decreased by 1.85 μg/m$^3$, 7.14 μg/m$^3$, 5.65 μg/m$^3$, and 15.98 μg/m$^3$, respectively, in Diqing, Naqu, Nagri, Haixi, and increased by 11.13 μg/m$^3$ in Shannan in 2016 relative to 2015, and increased by 20.85 μg/m$^3$ in Ngari in 2017 relative to 2016. As a result, all above reductions or increments in surface ozone level were mainly driven by anthropogenic emission. In contrast, surface ozone anomalies in Lhasa in 2017 and 2020, in Shigatse and Nyingchi in 2019 were mainly driven by meteorology.

Table S4 summarizes the inter-annual trends of surface ozone anomalies, meteorological and anthropogenic contributions from 2015 to 2020 in each city over the QTP. Except Guoluo, Qamdo and Lhasa which show decreasing trends, anthropogenic contributions in all other cities showed increasing trends from 2015 to 2020. With respect to meteorology contribution, Ngari, Naqu, Diqing and Haixi showed increasing trends from 2015 to 2020 and all other cities showed decreasing trends. The inter-annual trends of surface ozone anomalies in Ngari, Lhasa, Naqu, Qamdo, Diqing, Haixi and Guoluo can be attributed to anthropogenic emissions by 95.77%, 96.30%, 97.83%, 82.30%, 99.26%, and 87.85%, and meteorology by 4.23%, 3.70%, 2.17%, 3.19%, 0.74%, and 12.15%, respectively. The inter-annual trends of surface ozone in other cities were fully driven by anthropogenic emission, where the increasing inter-annual trends would have larger values if not for the favorable meteorological conditions. As a result, the inter-annual trends of surface ozone anomalies in all cities over the QTP were dominated by anthropogenic emission.

### 6.4 Discussions

Typically, all cities over the QTP are formed at flat valleys with surrounding mountains rising to more than 5.0 km a.s.l., and keep continuous expansion and development over time. Inhibited by surrounding mountains, regional dependent emissions and mountain peak-valley meteorological systems result in regional representation of surface zone level and their drivers on diurnal, seasonal, inter-annual scales.

Correlations between $O_{3,meteo}$ and each meteorological anomalies are summarized in Table S5. We find that all time scales of meteorology-driven surface ozone anomalies in each city are positively related with anomalies of temperature, planetary boundary layer height (PBLH), surface incoming shortwave flux (SWGDN), downward transport velocity at the PBLH (OMEGA), and tropopause height (TROPH). Among all these positive correlations, the correlations with PBLH, SWGDN and OMEGA in all cities are higher than those with TROPH. Since high temperature and SWGDN facilitate the formation of ozone via the increase in chemical reaction rates or biogenic emissions, the meteorology-driven surface ozone anomalies are consistent with the changes in temperature and SWGDN. Possible reasons for the ozone increases with the increase in PBLH include lower NO concentration at the urban surface due to the deep vertical mixing, which then limits ozone destruction and increases ozone concentrations (He et al., 2017), and more downward

transport of ozone from the free troposphere where the ozone concentration is higher than the near-surface concentration (Sun et al., 2009). Large OMEGA and high tropopause height also facilitate downward transport of stratospheric ozone, resulting in high surface ozone level. The QTP has been identified as a hot spot for stratospheric–tropospheric exchange (Cristofanelli et al., 2010;Škerlak et al., 2014) where the surface ozone is elevated from the baseline during the spring due to frequent stratospheric intrusions. Generally, surface ozone anomalies are negatively related with humidity, rainfall, and total cloud fraction in each city over the QTP. These wet meteorological conditions inhibit biogenic emissions, slow down ozone chemical production, and facilitate the ventilation of ozone and its precursors (Gong and Liao, 2019;Jiang et al., 2021;Lu et al., 2019a;Lu et al., 2019b;Ma et al., 2019), and therefore contribute to ozone decrease.

The $U_{10m}$ and $V_{10m}$ represent the metrics for evaluating the horizontal transport. In most of cities over QTP, noticeable ozone vs. horizontal wind correlations are observed, indicating that horizontal transport is an important contributor to surface ozone (Shen et al., 2014;Zhu et al., 2004). The QTP region, as a whole, is primarily regulated by the interplay of the Indian summer monsoon and the westerlies, and the atmospheric environment over QTP is heterogeneous. Mount Everest is representative of the Himalayas on the southern edge of the Tibetan Plateau and is close to South Asia where anthropogenic atmospheric pollution has been increasingly recognized as disturbing the high mountain regions (Decesari et al., 2010;Maione et al., 2011;Putero et al., 2014). In the northern QTP, including Xining, Haixi and Guoluo, is occasionally influenced by regional polluted air masses (Xue et al., 2011;Zhu et al., 2004), especially, the impacts of anthropogenic emissions from central and eastern China in the summer (Xue et al., 2011). For cities over the inland QTP, is distant from both South Asia and northwestern China; it has been found to be influenced by episodic long-range transport of air pollution from South Asia (Lüthi et al., 2015), evidenced by the study of aerosol and precipitation chemistry at these cities (Cong et al., 2010).

The monthly and annual averaged anthropogenic emissions of $NO_x$ and VOCs in each city over the QTP extracted from the MEIC (Multi-resolution Emission Inventory for China) inventory between 2015 to 2017 are presented in Table S6-S9. Major anthropogenic emissions in each city over the QTP are from transport sector and residential sector including burning emissions of coal, post-harvest crop residue, yak dung and religious incense (Chen et al., 2009;Kang et al., 2016;Kang et al., 2019;Li et al., 2017). Overall, both monthly and annual averaged anthropogenic contributions agree well with the changes of $NO_x$ and VOCs emissions in MEIC inventory (Table S6-S9).

## 7. Conclusions

In this study, we have investigated the evolutions, implications, and the drivers of surface ozone anomalies (defined as deviations of ozone levels relative to their seasonal means) between 2015 and 2020 in the urban areas over the QTP. Diurnal, seasonal, and inter annual variabilities of surface ozone in 12 cities over the QTP are analyzed. The average of surface ozone between 2015 and 2020 in each city over the QTP varied over (50.67 ±29.57) μg/m$^3$ to (90.38 ± 28.83) μg/m$^3$, and the median value varied over 53.00 μg/m$^3$ to 90.00 μg/m$^3$. Overall, diurnal cycle of surface ozone in each city over the QTP presents a unimodal pattern in all seasons. For all cities in all seasons, high levels of surface ozone occur in the daytime (9:00 to 20:00 LT) and low levels of surface ozone occur in the nighttime (21:00 to 08:00 LT). Seasonal cycles of surface ozone in most cities present a unimodal pattern with a seasonal peak occurs around March-July and a seasonal trough occurs around October-December. The inter-annual trends in surface ozone level from 2015 to 2020 over the QTP

spanned a large range of (−2.43 ± 0.56) μg/m$^3$·yr$^{-1}$ to (7.55 ± 1.61) μg/m$^3$·yr$^{-1}$, indicating a regional representation of each dataset.

We have established a RF regression model to describe the relationships between hourly surface ozone concentrations (response variables) and their potential driving factors (predictive variables) in the urban areas over the QTP. The RF model predictions and surface ozone measurements are in good agreement, showing high $R$ and low RMSE and MAE in each city over the QTP. Depending on cities, the $R$ values varied over 0.85 to 0.94, the RMSE over 10.24 to 17.55 μg/m$^3$, and MAE over 7.32 to 12.76 μg/m$^3$. The R, RMSE, and MAE are independent of city and surface ozone level. The results affirm that our model performs very well in predicting surface ozone levels and variabilities in each city over the QTP.

We have separated quantitatively the contributions of anthropogenic emission and meteorology to surface ozone anomalies by using the RF model based meteorological normalization method. Diurnal and seasonal surface ozone anomalies over the QTP were mainly driven by meteorology, and inter-annual anomalies were mainly driven by anthropogenic emission. Depending on region and measurement hour, diurnal surface ozone anomalies varied over -30.55 μg/m$^3$ to 34.01 μg/m$^3$ between 2015 and 2020, where meteorological and anthropogenic contributions varied over -20.08 μg/m$^3$ to 48.73 μg/m$^3$ and -27.18 μg/m$^3$ to 1.92 μg/m$^3$, respectively. Unfavorable meteorology driven 97% of surface ozone nonattainment events between 2015 and 2020 in the urban areas over the QTP. Monthly averaged surface ozone anomalies varied with much smaller amplitudes than their diurnal anomalies, where meteorological and anthropogenic contributions varied over 7.63 μg/m$^3$ to 55.61 μg/m$^3$ and 3.67 μg/m$^3$ to 35.28 μg/m$^3$ between 2015 and 2020, respectively. The inter-annual trends of surface ozone anomalies in Ngari, Lhasa, Naqu, Qamdo, Diqing, Haixi and Guoluo can be attributed to anthropogenic emissions by 95.77%, 96.30%, 97.83%, 82.30%, 99.26%, and 87.85%, and meteorology by 4.23%, 3.70%, 2.17%, 3.19%, 0.74%, and 12.15%, respectively. The inter-annual trends of surface ozone anomalies in other cities were fully driven by anthropogenic emission, where the increasing inter-annual trends would have larger values if not for the favorable meteorological conditions. This study can not only improve our knowledge with respect to spatiotemporal variability of surface ozone but also provides valuable implication for ozone mitigation over the QTP.

***Code and data availability.*** All other data are available on request of the corresponding author (Youwen Sun, ywsun@aiofm.ac.cn).

***Author contributions.*** HY designed the study and wrote the paper. YS supervised and revised this paper. JN, MP, and CL provided constructive comments.

***Competing interests.*** None.

***Acknowledgements.*** This work is jointly supported by the Youth Innovation Promotion Association, CAS (No.2019434) and the Sino-German Mobility programme (M-0036) funded by the National Natural Science Foundation of China (NSFC) and Deutsche Forschungsgemeinschaft (DFG).

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

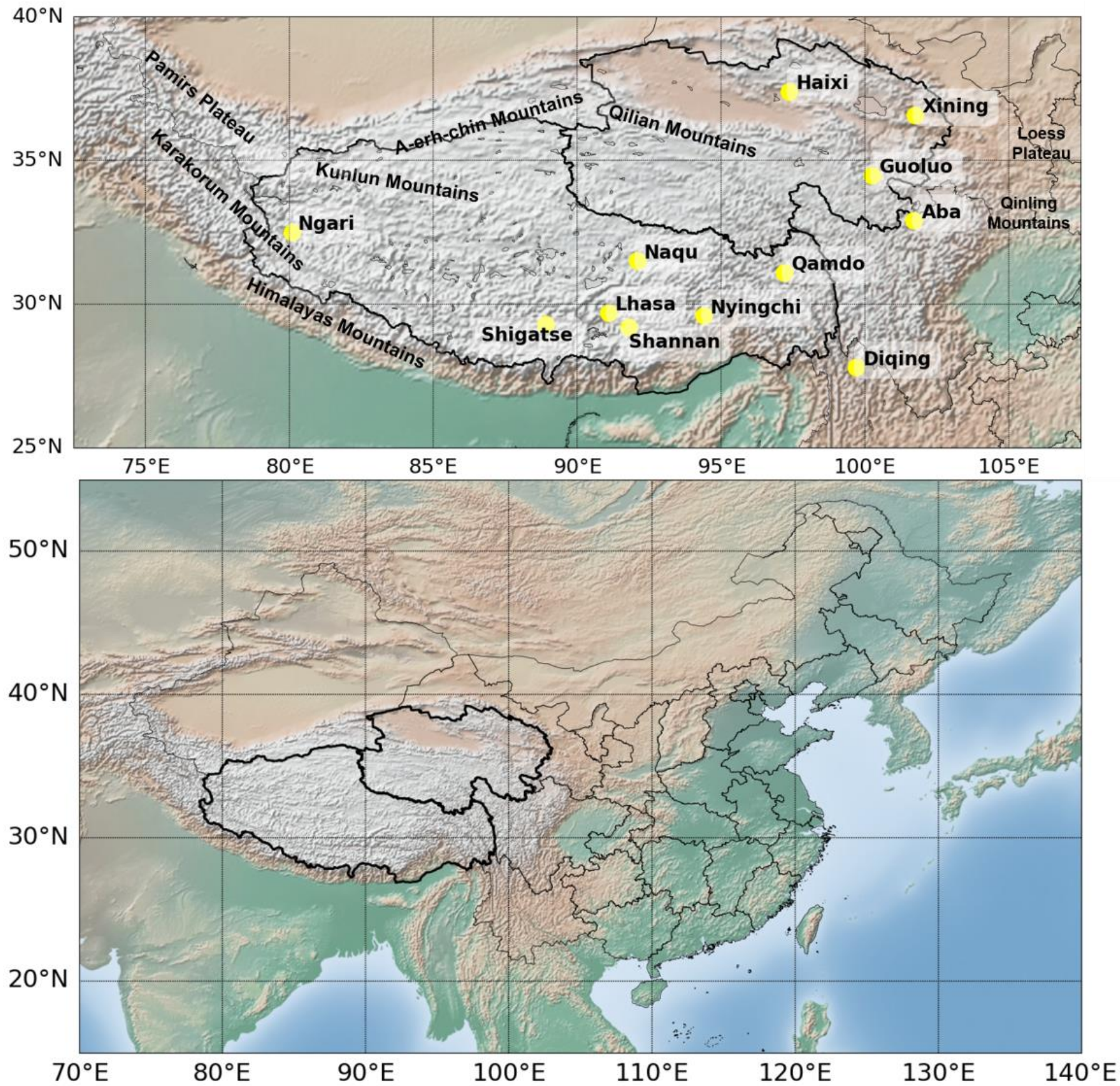

**Figure 1.** Geolocations of each city over the Qinghai-Tibet Plateau (QTP). The base map of the figure was created using the Basemap package in Python.

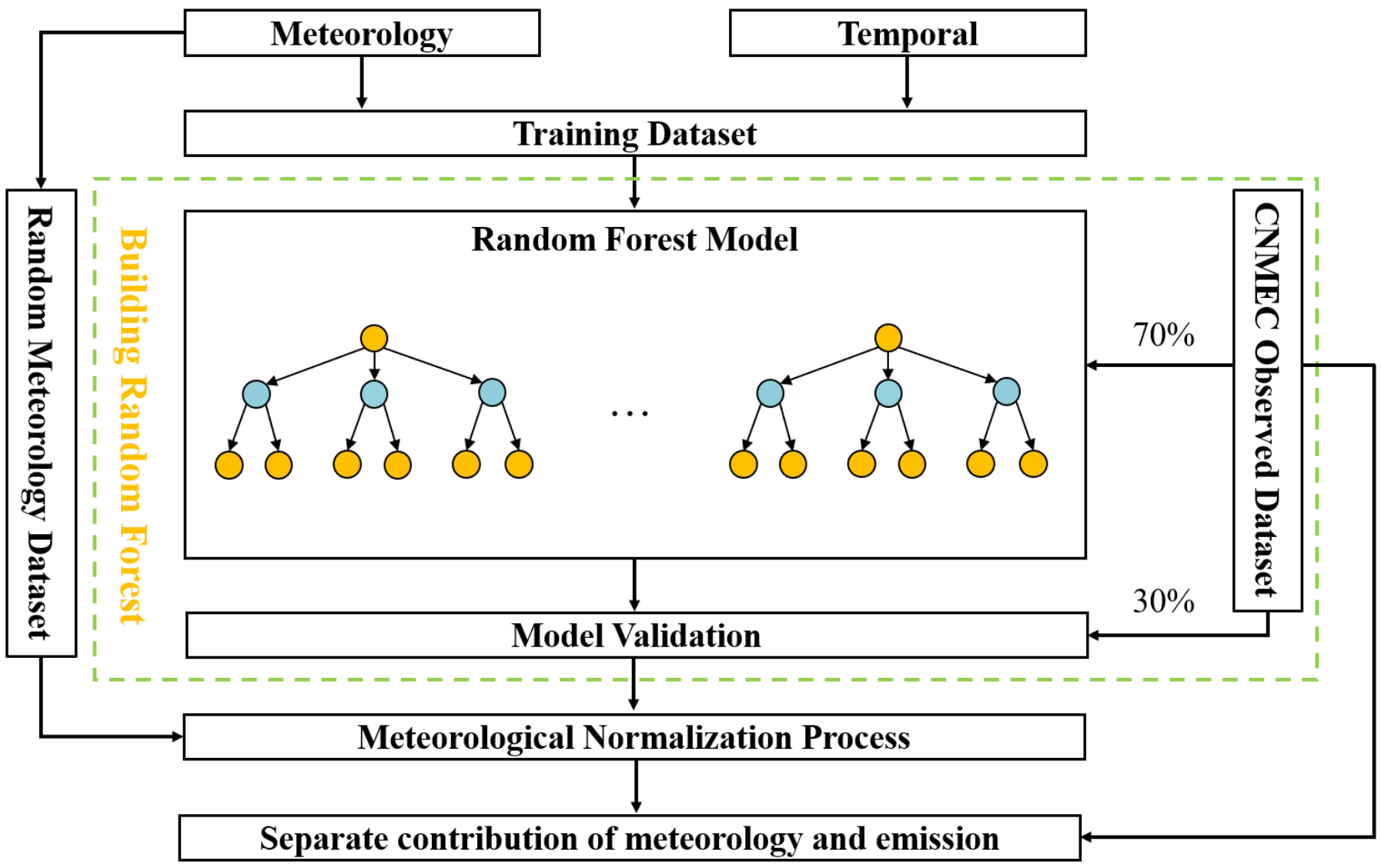

**Figure 2.** Flowchart for separation of meteorology and anthropologic contributions.

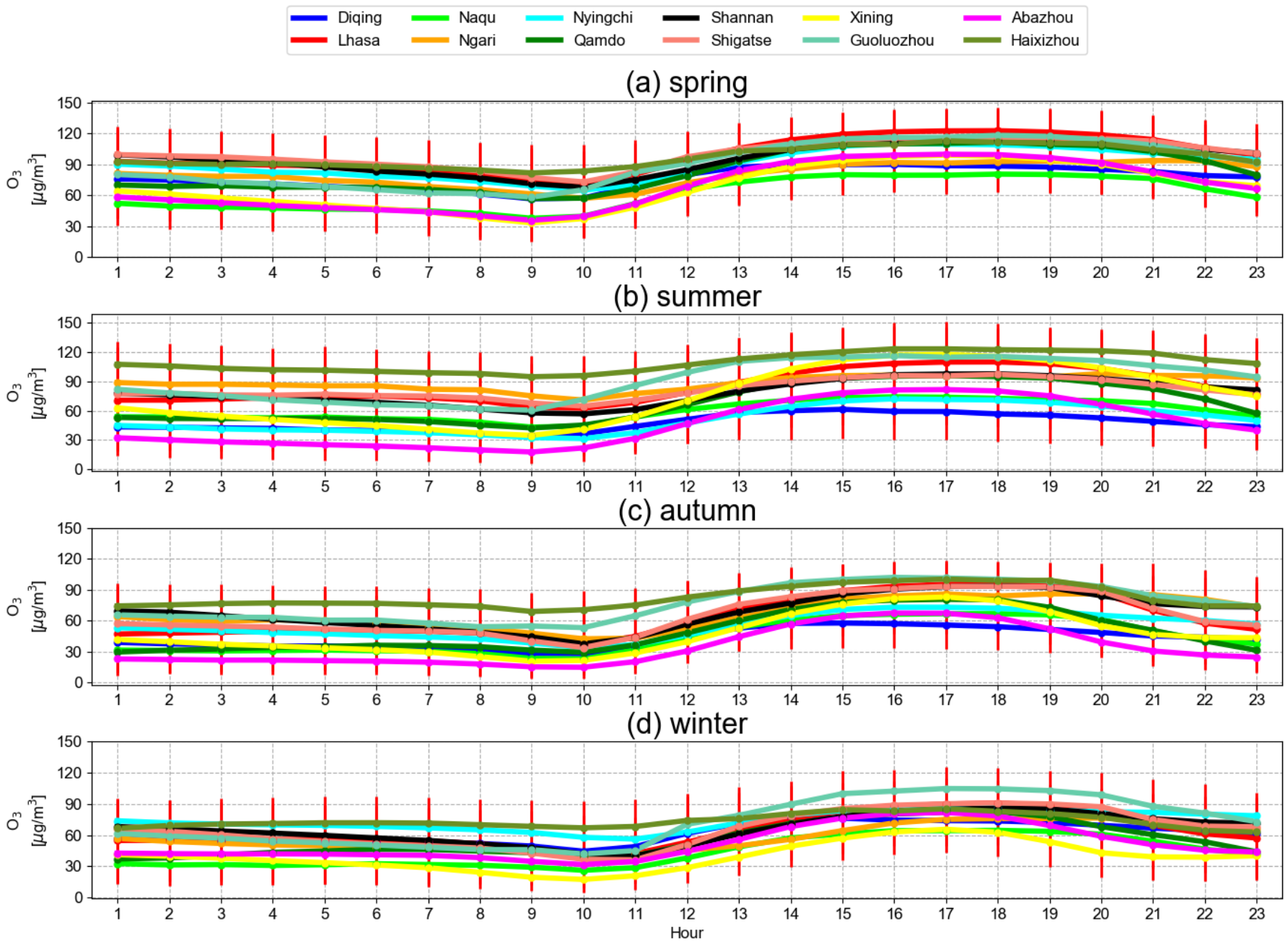

**Figure 3.** Diurnal cycle of surface ozone (units: μg/m$^3$) in each season and each city over the QTP. The vertical error bar is 1σ standard variation (STD) within that hour.

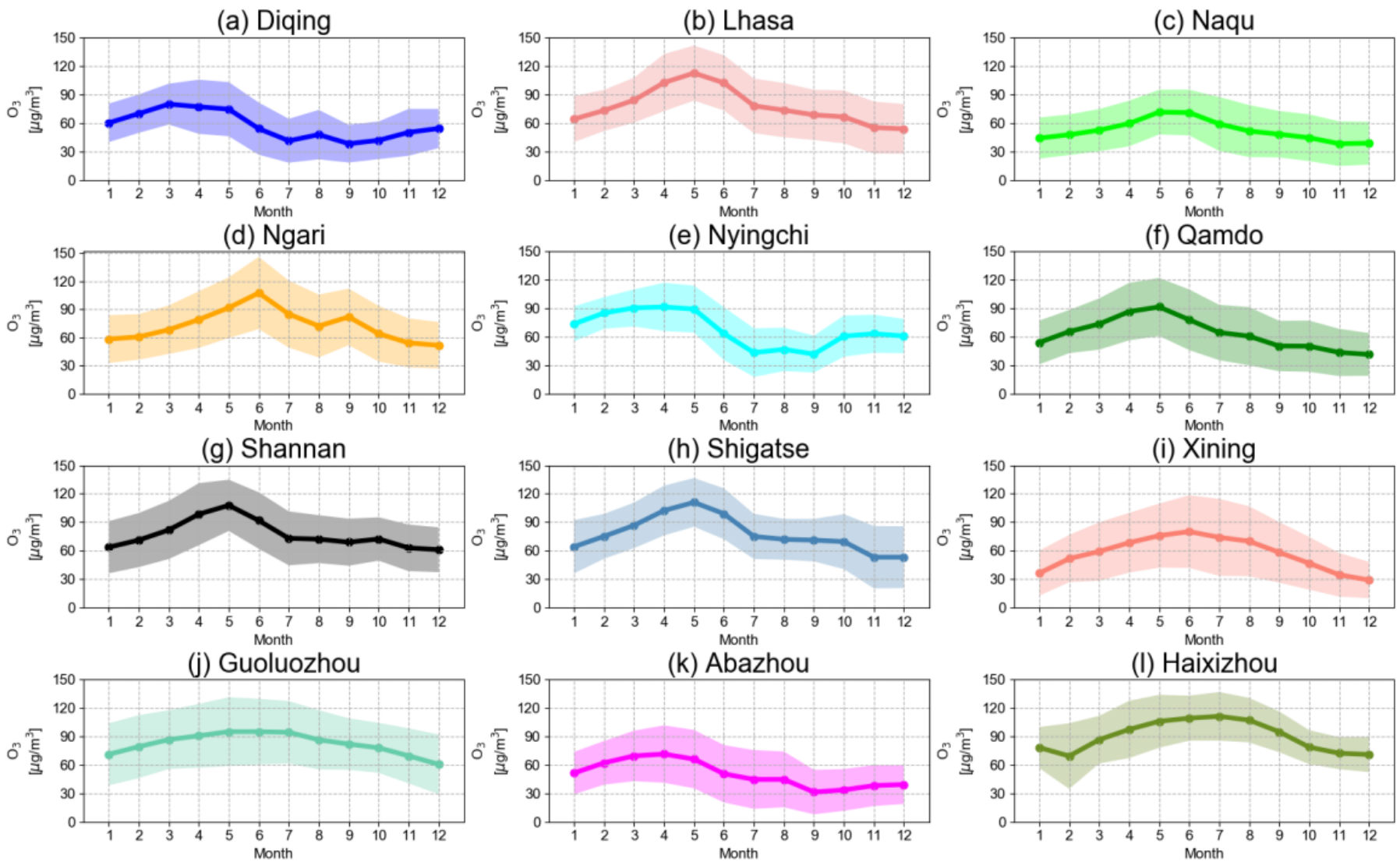

**Figure 4.** Monthly mean time series of surface ozone (units: μg/m$^3$) between 2015 and 2020 in each city over the QTP. The vertical error bar is 1σ standard variation (STD) within that month.

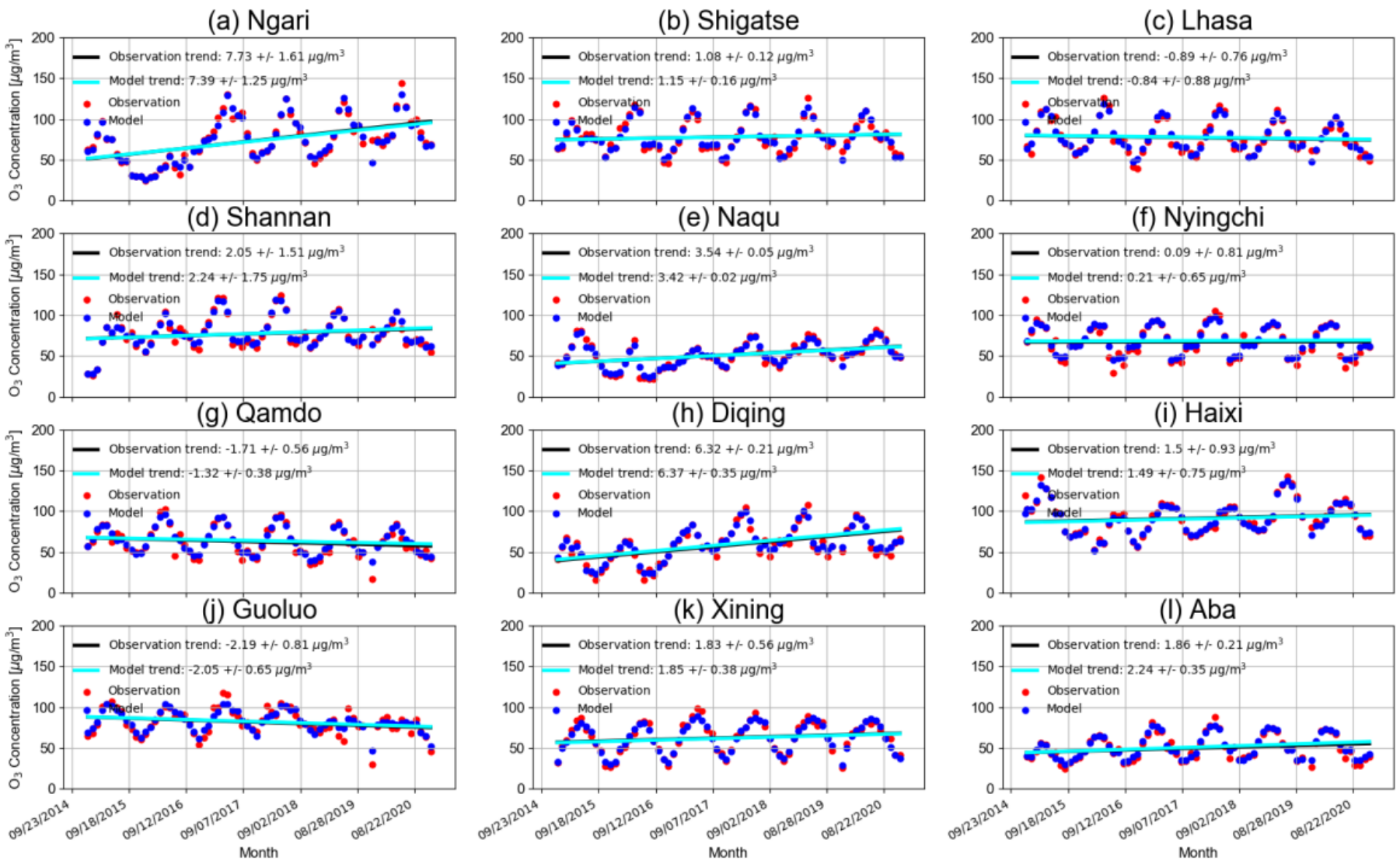

**Figure 5.** Inter-annual trends of surface ozone levels between 2015 and 2020 in the urban areas over the QTP. Blue dots are the monthly averaged surface ozone measurements. The seasonality and inter-annual variability in each city fitted by using a bootstrap resampling model with a second Fourier series (red dots) plus a linear function (black line) is also shown.

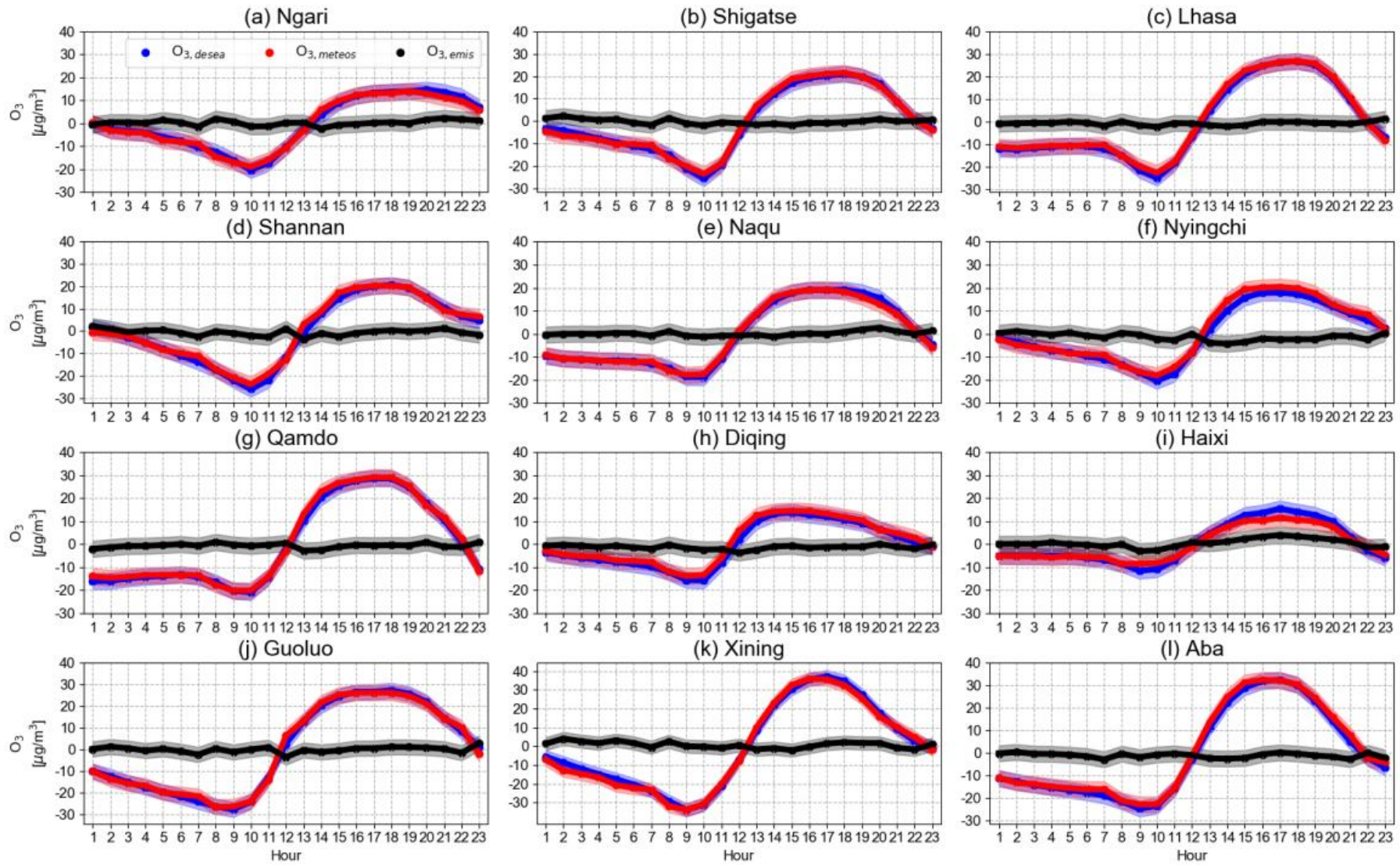

**Figure 6.** Diurnal cycles of surface ozone anomalies ($O_{3,anomalies}$, blue dots and lines) along with the meteorology-driven portions ($O_{3,meteo}$, red dots and lines) and the anthropogenic-driven portions ($O_{3,emis}$, black dots and lines) in each city over the QTP. Bold curves and the shadows are diurnal cycles and the 1σ standard variations, respectively.

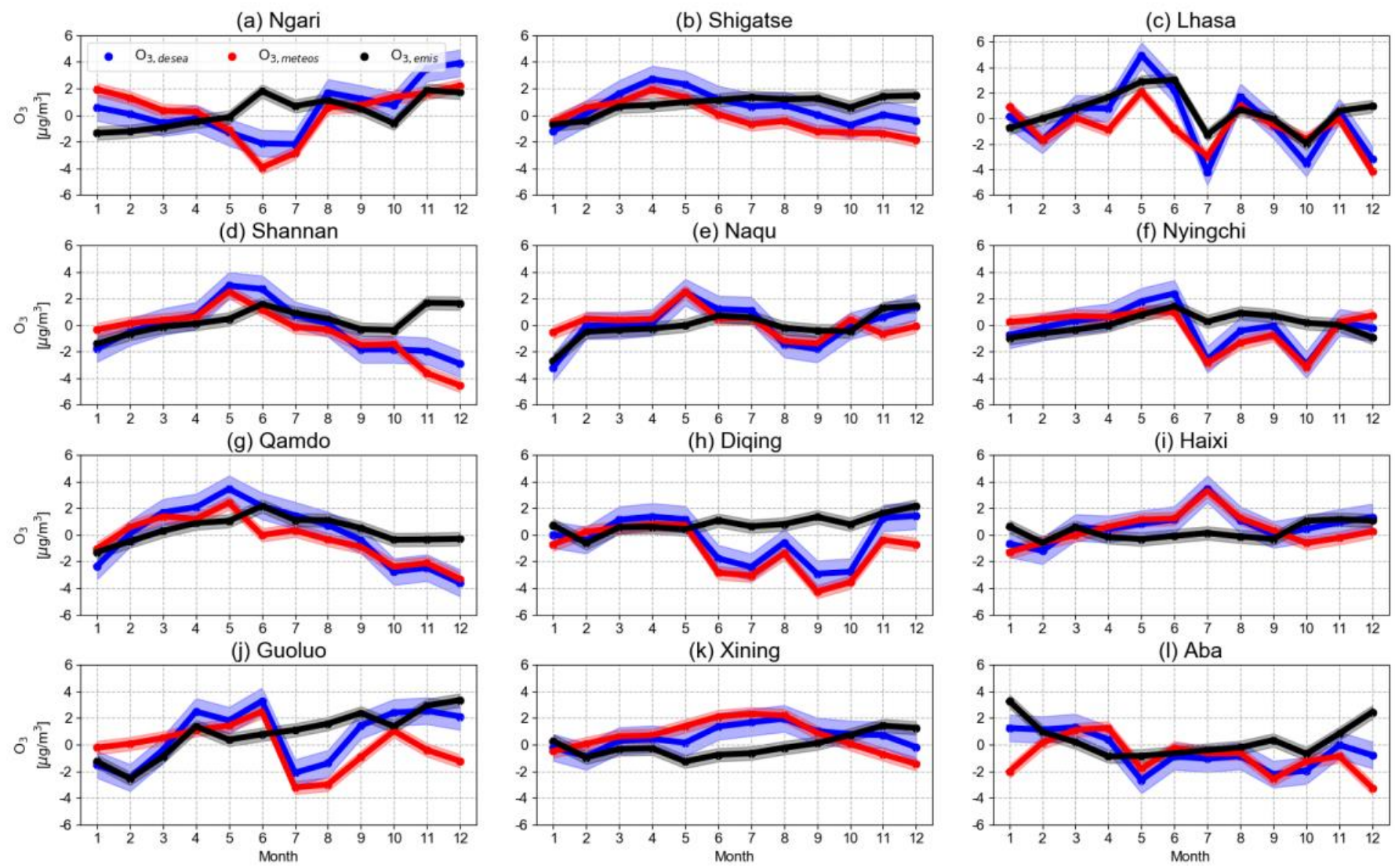

**Figure 7.** Seasonal cycles of surface ozone anomalies ($O_{3,anomalies}$, blue dots and lines) along with the meteorology-driven portions ($O_{3,meteo}$, red dots and lines) and the anthropogenic-driven portions ($O_{3,emis}$, black dots and lines) in each city over the QTP. Bold curves and the shadows are monthly mean values and the 1σ standard variations, respectively.

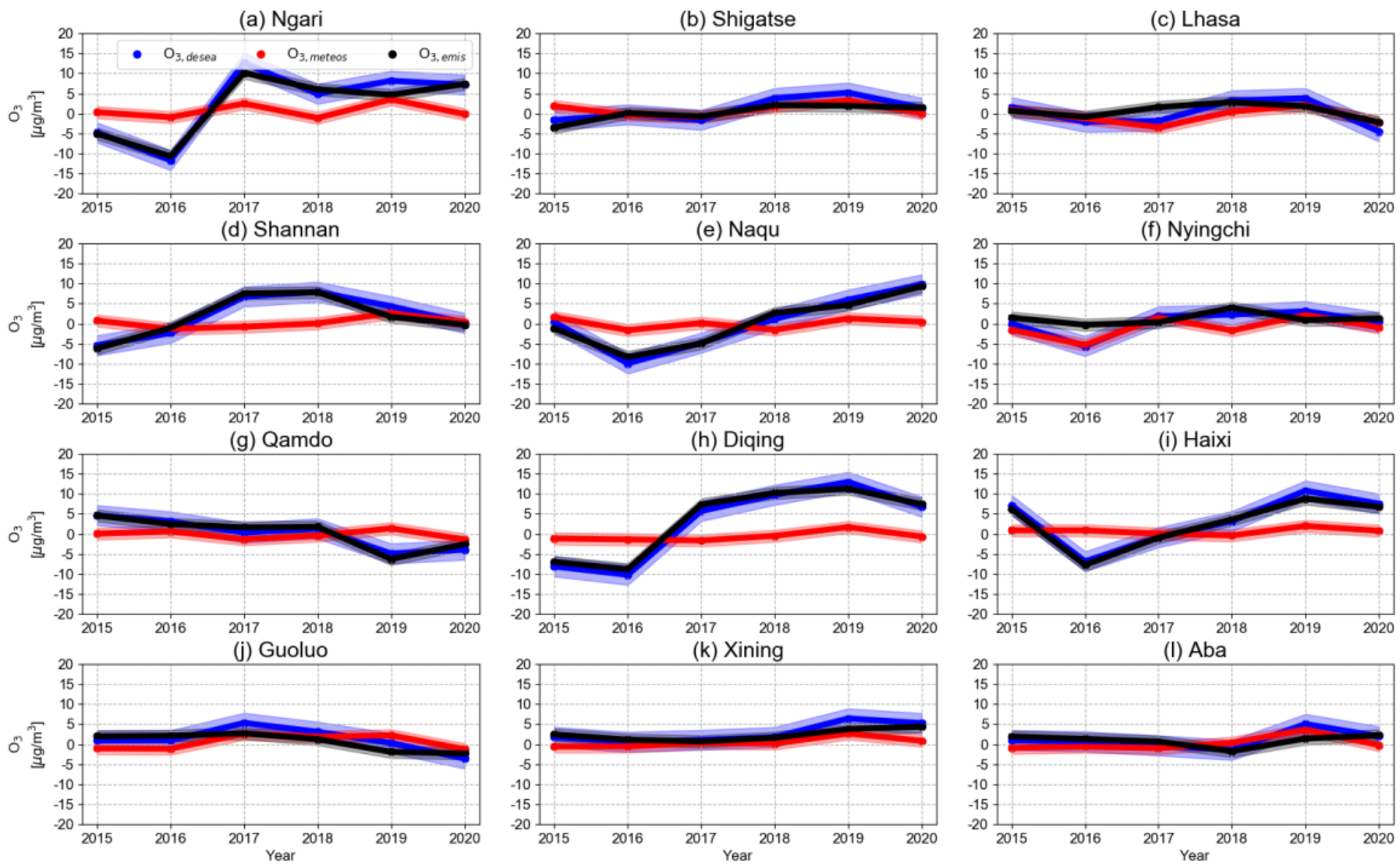

**Figure 8.** Annual mean surface ozone anomalies ($O_{3,anomalies}$, blue dots and lines) along with meteorology-driven portions ($O_{3,meteo}$, red dots and lines) and anthropogenic-driven portions ($O_{3,emis}$, black dots and lines) in each city over the QTP. Bold curves and the shadows are annual mean values and the 1σ standard variations, respectively.

**Table 1.** Geolocations of each city over the QTP. Population statistics are available from the 2020 nationwide population census issued by National Bureau of Statistics of China.

| Name | Latitude | Longitude | Number of site | Altitude (km) | Population (million) | Area (Thousand km$^2$) |
|---|---|---|---|---|---|---|
| Ngari | 32.5°N | 80.1°E | 2 | 4.5 | 0.12 | 345.0 |
| Shigatse | 29.3°N | 88.9°E | 3 | 4.0 | 0.80 | 182.0 |
| Lhasa | 29.7°N | 91.1°E | 6 | 3.7 | 0.87 | 31.7 |
| Shannan | 29.2°N | 91.8°E | 2 | 3.7 | 0.35 | 79.3 |
| Naqu | 31.5°N | 92.1°E | 3 | 4.5 | 0.50 | 430.0 |
| Nyingchi | 29.6°N | 94.4°E | 2 | 3.1 | 0.23 | 117.0 |
| Qamdo | 31.1°N | 97.2°E | 3 | 3.4 | 0.76 | 110.0 |
| Diqing | 27.8°N | 99.7°E | 2 | 3.5 | 0.39 | 23.9 |
| Haixi | 37.4°N | 97.4°E | 1 | 4.8 | 0.47 | 325.8 |
| Guoluo | 34.5°N | 100.3°E | 1 | 4.3 | 0.21 | 76.4 |
| Xining | 36.6°N | 101.7°E | 5 | 2.3 | 2.47 | 7.7 |
| Aba | 32.9°N | 101.7°E | 3 | 3.8 | 0.82 | 84.2 |

**Table 2.** List of predictive variables fed into the RF model.

| Parameters | Description | Unit |
|---|---|---|
| | Meteorological variables by MERRA-2 dataset | |
| $T_{surface}$ | Surface air temperature | °C |
| $U_{10m}$ | zonal wind at 10 m height | m/s |
| $V_{10m}$ | meridional wind at 10 m height | m/s |
| PBLH | Planetary boundary layer height | m |
| CLDT | Total cloud area fraction | unitless |
| PRECTOT | Total Precipitation | kg·m$^2$/s |
| OMEGA | Vertical pressure velocity at PBLH | Pa/s |
| SWGDN | Surface incoming shortwave flux | W/m$^2$ |
| QV | Specific humidity at 2 m height | kg/kg |
| TROPT | Tropospheric layer pressure | Pa |
| | Time information | |
| Year | Year since 2015 | / |
| Month | Month of the year | / |
| day | Day of the month | / |
| Hour | Hour of the day | / |