# Peer review of "Quantifying the drivers of surface ozone anomalies in the urban"

_Atmospheric Chemistry and Physics, 2022_

## Author Comment (AC1)

**Response to Referee #1**:

Thanks very much for your comments, suggestions and recommendation with respect to improve this paper. The response to all your comments are listed below.

This paper investigate the mechanism of short-term surface ozone anomalies in the urban areas over the Qinghai-Tibet Plateau. The topic and presentation are fine in general. A revision according the following comments should be provided.

**Response:** All your comments listed below have been addressed. Please check the point by point response as follows.

**Comment [1-1]:** The description for producing meteorological normalized concentration in Section 3.3 is quite vague. For example, (1) how the final result be sensitive to the four-week period window? If it is chosen to reflect the seasonal variability, is it really considered to be superior/useful than traditional deseasonalization methods? (2) p6, l17, "This selection process was repeated 1000 times to generate a final input dataset." How is the final input being generated exactly? By using sample mean, median, or anything else? Since the figure results presented in this study are not as variable as I expected from the main text, I am not quite convinced that a random resampling method would lead to such smooth results. A time series plot of original data for each station is desired.

**Response:** Thanks for your suggestions. In the revised version, we believe the combination of section 3.2 and 3.3 can explain the meteorological normalized concentration clearly. (1) The method and the selection of "the four-week period window" in this study follow those of Vu et al. (2019) and Shi et al. (2021). We resample observed weather data within a four-week period for a longer period (1980 to 2020) rather than only the study period, which normalizes the impact of weather variations but not the seasonal variations. This method enables us to investigate the seasonality of weather normalized concentrations (Vu et al., 2019;Shi et al., 2021). The meteorological normalized method is more useful than traditional deseasonalization methods since it is able to separate the contributions of meteorology and anthropologic emission to surface ozone anomalies. (2) The 1000 predicted concentrations were then averaged to calculate the final weather normalized concentration for that particular hour, day, and year. For each measurement, we resample the observed weather data within a four-week period for a longer period (1980 to 2020) 1000 times so that all kinds of weather conditions around the measurement time have been considered in the model predictions. The purpose of this process is to collect enough data and eliminate the influence of abnormal meteorological conditions, and get concentrations under the averaged meteorological conditions. We have added the content to Page 6, Line 34-36. Because the weather normalized concentrations are the averaged values, it is normal for the time series by random resampling method to be smooth. Similarly, the results by Vu et al. (2019) and Shi et al. (2021) are also smooth. In the revised version, the time series plots of original data for each city are presented in Supplement Figure S3 (i.e., Figure R1 in this file).

[Figure]

**Figure R1** Time series of surface ozone observations and meteorological normalization data in each city over the QTP region.

**Comment [1-2]:** A significant portion of this study is devoted to the discussion of ozone extreme values. To provide a more systematic discussion, and to facilitate better communication, I suggest the authors should quantitatively work on the percentile variation instead (e.g. the 5th and 95th), as suggested by following references:

Cooper, O. R., Gao, R. S., Tarasick, D., Leblanc, T., & Sweeney, C. (2012). Long-term ozone trends at rural ozone monitoring sites across the United States, 1990–2010. Journal of Geophysical Research: Atmospheres, 117(D22).

Munir, S., Chen, H., & Ropkins, K. (2012). Modelling the impact of road traffic on ground level ozone concentration using a quantile regression approach. Atmospheric environment, 60, 283-291.

Chang, K. L., Schultz, M. G., Lan, X., McClure-Begley, A., Petropavlovskikh, I., Xu, X., & Ziemke, J. R. (2021). Trend detection of atmospheric time series: Incorporating appropriate uncertainty estimates and handling extreme events. Elem Sci Anth, 9(1), 00035.

Wells, B., Dolwick, P., Eder, B., Evangelista, M., Foley, K., Mannshardt, E., ... & Weishampel, A. (2021). Improved estimation of trends in US ozone concentrations adjusted for interannual variability in meteorological conditions. Atmospheric Environment, 248, 118234.

**Response:** Thanks for your suggestions. In revised version, we have presented the percentile variation of surface ozone concentration (units: $\mu g/m^3$) in each city over the QTP from 2015 to 2020 in Figure S2 (i.e., Figure R2 in this file). The percentile variation modes of surface ozone concentration in all cities over the QTP are similar. In this study, only mean plus standard variance of surface ozone concentration rather than its percentile variation in each city was investigated. This prevailing method has been used in a number of studies to describe the variabilities of atmospheric compositions over the QTP, such as Li et al. (2020), Liu et al. (2021), Ma et al. (2020), Xu et al.

(2018), Xu et al. (2016), Yin et al. (2019), and Yin et al. (2017). We have added these contents to Page 7, Line 38-43 and Page 8, Line 1-2. Please check it. The method (mean + standard variance) can also well reflect the trends and variabilities of ozone, and can also provide a more systematic discussion and communication.

[Figure]

**Figure R2.** The percentile variation of surface ozone concentration (units: μg/m$^3$) in each city over the QTP from 2015 to 2020.

Minor suggestions:

**Comment [1-3]:** p4, l6-7, the data quality control procedures should be briefly stated.

**Response:** Thanks for your reminder. The filter criteria can be summarized as follows. Hourly observed data points were transformed into Z scores (Same as the uniformized process and will be refer below), and then, the observed data were removed if the corresponding $Z_i$ met one of the following conditions: (1) $Z_i$ is larger or smaller than the previous one ($Z_{i-1}$) by 9 ($|Z_i - Z_{i-1}| > 9$), (2) The absolute value of $Z_i$ is greater than 4 ($|Z_i| > 4$), or (3) the ratio of the Z value to the third-order center moving average is greater than 2 ($\frac{3Z_i}{Z_{i-1}+Z_i+Z_{i+1}} > 2$). The uniformized process are presented as follows:

$$\mathbf{z}_k = \frac{\mathbf{x}_k - \mathbf{u}_k}{\mathbf{\sigma}_k} \tag{1}$$

where $\mathbf{u}_k$ and $\mathbf{\sigma}_k$ are the average and 1σ standard deviation (STD) of $\mathbf{x}_k$, and $\mathbf{z}_k$ is the pre-processed value for parameter $\mathbf{x}_k$. We have added the data quality control procedures in Page 4, Line 22-30. Please check it.

**Comment [1-4]:** p6, l1-2, this part seems to come from nowhere.

**Response:** Thanks for your reminder. We have moved this part to Page 4, Line 22-30. Please check it.

**Comment [1-5]:** p8, l4, Yin et al. (2017)

**Response:** Thanks for your reminder. We have corrected this mistake. Please check it.

**Comment [1-6]:** p10,l25, what does it mean for "seasonal cycles of surface ozone anomalies"? Should the anomaly is deseasonalized already in Eq (5)? If it refers to remaining seasonality variation, can it imply that the methodology in Eq (5) is not appropriate?

**Response:** We calculate surface ozone anomalies ($O_{3,anomalies}$) in each city over the QTP by subtracting their seasonal mean values ($O_{3,mean}$) from all hourly surface ozone measurements ($O_{3,individual}$). We then discussed the surface ozone anomalies and separated the contributions of anthropogenic emissions and meteorological conditions on different time scales. For example, when we discuss seasonal cycles of surface ozone anomalies, we calculate monthly mean values of surface ozone anomalies, and investigate the month-to-month variabilities of the anomalies throughout the year. Similarly, for diurnal scale, we calculate hourly mean values of surface ozone anomalies, and investigate the hour-to-hour variabilities of the anomalies throughout the day. As a result, we just summarize the anomalies on seasonal scale and it doesn't mean that the methodology in Equation (5) is not appropriate. The purpose of Equation (5) is only to find surface ozone anomalies.

**Reference**

Li, R., Zhao, Y. L., Zhou, W. H., Meng, Y., Zhang, Z. Y., and Fu, H. B.: Developing a novel hybrid model for the estimation of surface 8 h ozone ($O_3$) across the remote Tibetan Plateau during 2005-2018, Atmos Chem Phys, 20, 6159-6175, 2020.

Liu, S., Fang, S., Liu, P., Liang, M., Guo, M., and Feng, Z.: Measurement report: Changing characteristics of atmospheric $CH_4$ in the Tibetan Plateau: records from 1994 to 2019 at the Mount Waliguan station, Atmos. Chem. Phys., 21, 393-413, 10.5194/acp-21-393-2021, 2021.

Ma, J., Dörner, S., Donner, S., Jin, J., Cheng, S., Guo, J., Zhang, Z., Wang, J., Liu, P., Zhang, G., Pukite, J., Lampel, J., and Wagner, T.: MAX-DOAS measurements of $NO_2$, $SO_2$, HCHO, and BrO at the Mt. Waliguan WMO GAW global baseline station in the Tibetan Plateau, Atmos. Chem. Phys., 20, 6973-6990, 10.5194/acp-20-6973-2020, 2020.

Shi, Z. B., Song, C. B., Liu, B. W., Lu, G. D., Xu, J. S., Vu, T. V., Elliott, R. J. R., Li, W. J., Bloss, W. J., and Harrison, R. M.: Abrupt but smaller than expected changes in surface air quality attributable to COVID-19 lockdowns, Sci Adv, 7, 2021.

Vu, T. V., Shi, Z. B., Cheng, J., Zhang, Q., He, K. B., Wang, S. X., and Harrison, R. M.: Assessing the impact of clean air action on air quality trends in Beijing using a machine learning technique, Atmos Chem Phys, 19, 11303-11314, 2019.

Xu, W., Xu, X., Lin, M., Lin, W., Tarasick, D., Tang, J., Ma, J., and Zheng, X.: Long-term trends of surface ozone and its influencing factors at the Mt Waliguan GAW station, China – Part 2: The roles of anthropogenic emissions and climate variability, Atmos. Chem. Phys., 18, 773-798, 10.5194/acp-18-773-2018, 2018.

Xu, W. Y., Lin, W. L., Xu, X. B., Tang, J., Huang, J. Q., Wu, H., and Zhang, X. C.: Long-term trends of surface ozone and its influencing factors at the Mt Waliguan GAW station, China - Part 1: Overall trends and characteristics, Atmos Chem Phys, 16, 6191-6205, 2016.

Yin, X. F., Kang, S. C., de Foy, B., Cong, Z. Y., Luo, J. L., Zhang, L., Ma, Y. M., Zhang, G. S., Rupakheti, D., and Zhang, Q. G.: Surface ozone at Nam Co in the inland Tibetan Plateau: variation, synthesis

comparison and regional representativeness, Atmos Chem Phys, 17, 11293-11311, 2017.

Yin, X. F., de Foy, B., Wu, K. P., Feng, C., Kang, S. C., and Zhang, Q. G.: Gaseous and particulate pollutants in Lhasa, Tibet during 2013-2017: Spatial variability, temporal variations and implications, Environmental Pollution, 253, 68-77, 2019.

---

## Author Comment (AC2)

**Response to Referee #2:**

Thanks very much for your comments, suggestions and recommendation with respect to improve this paper. The response to all your comments are listed below.

This study examined surface ozone variation over the QTP regions and quantified the role of anthropogenic emissions and meteorology in ozone changes from daily scale to multiyear scale using the random forest model. I think the topic of this study is within the scope of ACP journal. The dataset and method applied here are reasonable. However, I feel the results are overall descriptive and more in-depth analysis should be done to improve the current manuscript.

**Response:** All your comments listed below have been addressed. Please check the point by point response as follows.

**Comment [2-1]:** The authors said that the implication of this study is for ozone control. However, the mean ozone level over QTP is far behind the national AQ standard. Is ozone an air pollution issue over QTP? They summarized the ozone nonattainment days but listed them as Table S1 in the supplementary. This table should be move into the main text.

**Response:** We have moved Table S1 to the main text as Table 3. Please check it. Indeed, the mean ozone level over QTP is far behind the national AQ standard but we observed ozone nonattainment events over the QTP. Due to its unique features of landform, ecosystem and monsoon circulation pattern, the QTP has been regarded as a sensitive region to anthropogenic impact, and is referred to as an important indicator of regional and global climate change. The exogenous and local atmospheric pollutants are potential to accelerate the melting of glaciers, damage air quality, water sources, and grasslands, and threaten climate on regional and global scales. This study can separate quantitatively the contributions of anthropogenic emission and meteorology to surface ozone anomalies by using the RF model based meteorological normalization method. Separation of anthropogenic and meteorological drivers is very important since it conveys us exactly which processes drive the observed ozone anomaly and therefore right conclusions can be made on whether an emission mitigation policy is effective. This study can not only improve our knowledge with respect to spatiotemporal variability of surface ozone but also provides valuable implication for ozone mitigation over the QTP.

**Comment [2-2]:** The authors quantified the anthropogenic and meteorological contributions to ozone changes at different time scale, but they failed to explain them further. In Section 6.4, they only discussed the anthropogenic or meteorological roles generally. But it is important to know if it works for all time scales. Or the authors could focus on some time scale to discuss.

**Response:** In revised version, we quantified the anthropogenic and meteorological contributions to ozone changes at different time scale, and they presented in-depth analysis. We first present descriptively the contributions of anthropogenic emission and meteorology to surface ozone anomalies over the QTP in section 6.1 to 6.3, where statistics on different time scales were summarized. We then present in-depth analysis of each driver in section 6.4. In section 6.4, we not only discussed the mechanisms that work for all time scales but also discussed the mechanisms that work for a specific time scale, e.g., the specific ozone nonattainment events. Please check section 6.4 for details.

**Comment [2-3]:** P9L37: I don't understand why it is needed to quantify anthropogenic and meteorological contribution to ozone changes at the diurnal scale.

**Response:** For the investigation on diurnal scale, we calculate hourly mean values of surface ozone anomalies, and investigate the hour-to-hour variabilities of the anomalies throughout the day. This diurnal scale investigation allows us to determine the drivers of daily surface ozone nonattainment events or specific ozone nonattainment events. As a result, we have concluded in Section 6.3, "Exceptional meteorology driven 97% of surface ozone nonattainment events from 2015 to 2020 in the urban areas over the QTP. For the meteorology-dominated surface ozone nonattainment events, meteorological and anthropogenic contributions varied over 32.85 $\mu g/m^3$ to 55.61 $\mu g/m^3$ and 3.67 $\mu g/m^3$ to 7.23 $\mu g/m^3$, respectively. For the anthropogenic-dominated surface ozone nonattainment events, meteorological and anthropogenic contributions varied over 7.63 $\mu g/m^3$ to 10.53 $\mu g/m^3$ and 15.63 $\mu g/m^3$ to 35.28 $\mu g/m^3$, respectively."

**Comment [2-4]:** P4L27-28: Has MERRA2 data verified by surface measurements over QTP?

**Response:** Wang and Zeng (2012) has compared the MERRA2 products with surface measurements at 63 weather stations over the QTP region from the Chinese Meteorological Administration (CMA). Xie et al. (2017) also compared the meteorological parameters provide by MERRA2 data and CMA observations. These results demonstrate the accuracy of MERRA2 data. We have added these references to corresponding sentences (Page 4, Line 39). Please check it.

**Comment [2-5]:** P6L18-19: change "zone" to "ozone". Please check over through the text.

**Response:** Thanks for your reminder. We have check and revised these mistakes. Please check it.

**Comment [2-6]:** P10L19: Could you specify the dominant meteorological variables responsible for ozone nonattainment events?

**Response:** In order to determine which specific meteorological variables responsible for the meteorology-dominated ozone nonattainment events over the QTP, we have investigated the correlations between each meteorological variable and ozone anomalies in each city during the ozone nonattainment days. As tabulated in Table S8 (i.e., Table R1 in this file), temperature is the dominant meteorological variable responsible for the meteorology-dominated ozone nonattainment events, especially in Shigatse, Lhasa, Shannan, Haixi and Guoluo. In addition, the OMEGA is also an important meteorological variable in most cities, especially in Guoluo where the correlation is up to 0.69. For other meteorological variables, winds ($U_{10m}$, $V_{10m}$) and TROPH also have noticeable contributions to some ozone nonattainment events. We have added these contents to Page 13, Line 12-19. Please check it.

**Table R1** The correlations between each meteorological variable and ozone anomalies in each city over the QTP region during ozone nonattainment events.

| City | Correlations | | | | | | | | | |
|---|---|---|---|---|---|---|---|---|---|---|
| | $T_{surface}$ | $U_{10m}$ | $V_{10m}$ | PBLH | TCC | Rain | Omega | SWGDN | $RH_{2m}$ | TROPH |
| Ngari | 0.57 | -0.45 | -0.13 | 0.09 | 0.35 | 0.38 | 0.32 | -0.25 | -0.02 | 0.16 |
| Shigatse | 0.69 | 0.38 | -0.02 | 0.29 | -0.13 | -0.37 | 0.31 | -0.37 | -0.36 | 0.23 |
| Lhasa | 0.51 | 0.35 | -0.12 | 0.34 | -0.15 | -0.39 | 0.35 | 0.02 | -0.36 | 0.18 |
| Shannan | 0.67 | -0.22 | -0.25 | 0.02 | 0.22 | 0.14 | 0.25 | -0.04 | -0.11 | 0.32 |
| Naqu | NA[1] | NA | NA | NA | NA | NA | NA | NA | NA | NA |

| | | | | | | | | | | |
|---|---|---|---|---|---|---|---|---|---|---|
| Nyingchi | NA | NA | NA | NA | NA | NA | NA | NA | NA | NA |
| Qamdo | NA | NA | NA | NA | NA | NA | NA | NA | NA | NA |
| Diqing | NA | NA | NA | NA | NA | NA | NA | NA | NA | NA |
| Haixi | 0.83 | 0.28 | -0.08 | 0.40 | 0.10 | 0.23 | 0.22 | -0.77 | -0.38 | 0.30 |
| Guoluo | 0.52 | -0.76 | -0.34 | 0.15 | 0.39 | -0.12 | 0.69 | 0.45 | -0.34 | 0.33 |
| Xining | 0.69 | -0.20 | -0.37 | 0.34 | 0.35 | 0.45 | 0.36 | 0.08 | -0.20 | 0.31 |
| Aba | NA | NA | NA | NA | NA | NA | NA | NA | NA | NA |

[1] In these cities, there is no ozone nonattainment events during 2015 to 2020, expected Qamdo. In Qamdo, the ozone nonattainment events are only in 2 days. Therefore, we cannot calculate the correlations between each meteorological variable and ozone anomalies in these cities.

**Comment [2-7]:** P11L1: This section talks about ozone changes over multiyear time frame.
**Response:** We have modified the title of this section to "Multi-year scale". Please check it.

**Comment [2-8]:** P11L12-13: It is not clear how these ozone changes were driven by anthropogenic emissions.
**Response:** In the revised version, we used the annual averaged anthropogenic emissions of $NO_x$ and VOCs in each city over the QTP region extracted from the MEIC (Multi-resolution Emission Inventory for China) inventory between 2015 to 2017 to explain the ozone changes. We can find that the emissions of $NO_x$ and VOCs have been decreased in Diqing, Naqu, Nagri in 2016 relative to 2015. The reduction of $NO_x$ and VOCs emissions jointly drives the changes of ozone in these cities. Although VOCs emissions increased in Haixi during 2015 to 2016, $NO_x$ emissions have significantly decreased by 6.82 t. As a result, the decreases in ozone in 2016 relative to 2015 in Haixi are attributed to the significant reduction in $NO_x$ emissions. As the MEIC inventory is only updated to 2017, the inventory of ozone precursors for 2018-2020 is currently unavailable. However, the inventory from 2015 to 2017 can also roughly explain the multi-year scale change of ozone. We have added these contents in Page 13, Line 20-23 and Line 25-30. Please check it.

**Comment [2-9]:** P12L30-31: Please make it clear how ozone changes are consistent with emission changes.
**Response:** We presented the correlations between the monthly averaged anthropogenic contributions and $NO_x$ and VOCs emissions by MEIC inventory. The tables have added in supplement (Table S13, i.e., Table R2 in this file). The correlations of the monthly averaged anthropogenic contributions against anthropogenic $NO_x$ and VOCs emissions are in the range of 0.35-0.81 and 0.33-0.83, respectively. For the annual averaged statistics, the correlations against $NO_x$ and VOCs emissions are in the range of 0.15-0.94 (expect for Nyingchi and Diqing), and 0.34-0.98 (expect for Haixi), respectively. For all cities except Shannan, Qamdo and Haixi, both the $NO_x$ and VOCs emissions are consistent with the anthropogenic contributions. While only $NO_x$ emissions in Qamdo and Haixi and VOCs emissions in Shannan are consistent with anthropogenic contributions. In general, the changes of $NO_x$ and VOCs emissions in MEIC inventory are able to explain the variabilities of both monthly and annual averaged anthropogenic contributions. We have added these contents in Page 13, Line 31-39. Please check it.
**Table R2** The correlations between the monthly averaged anthropogenic contributions and $NO_x$ and VOCs emissions by MEIC inventory.

| City | $R_{NO_x-month}$ | $R_{VOC-month}$ | $R_{NO_x-year}$ | $R_{VOC-year}$ |
|------|------------------|-----------------|-----------------|-----------------|
| Ngari | 0.74 | 0.62 | 0.15 | 0.88 |
| Shigatse | 0.58 | 0.61 | 0.56 | 0.62 |
| Lhasa | 0.43 | 0.33 | 0.28 | 0.34 |
| Shannan | 0.35 | 0.65 | 0.94 | 0.98 |
| Naqu | 0.56 | 0.66 | 0.78 | 0.82 |
| Nyingchi | 0.65 | 0.61 | -0.84 | 0.54 |
| Qamdo | 0.75 | 0.35 | 0.82 | 0.83 |
| Diqing | 0.66 | 0.55 | -0.29 | 0.77 |
| Haixi | 0.81 | 0.39 | 0.92 | -0.36 |
| Guoluo | 0.74 | 0.71 | 0.93 | 0.92 |
| Xining | 0.55 | 0.83 | 0.91 | 0.90 |
| Aba | 0.77 | 0.67 | 0.87 | 0.89 |

**Reference**

Wang, A., and Zeng, X.: Evaluation of multireanalysis products with in situ observations over the Tibetan Plateau, Journal of Geophysical Research: Atmospheres, 117, https://doi.org/10.1029/2011JD016553, 2012.

Xie, Z., Hu, Z., Gu, L., Sun, G., Du, Y., and Yan, X.: Meteorological Forcing Datasets for Blowing Snow Modeling on the Tibetan Plateau: Evaluation and Intercomparison, Journal of Hydrometeorology, 18, 2761-2780, 10.1175/JHM-D-17-0075.1, 2017.